# Learning curves for itinerant nurses to master the operation skill of Ti-robot-assisted spinal surgery equipment by CUSUM analysis: A pilot study

Yichao Yao[1], Huiyue Wang[2], Qi Zhang[1]*, Haimao Teng[1], Hui Qi[3], Qian Zhang[3]

1 Department of Operating Room, Baoding First Central Hospital, Baoding, China, 2 Department of Anesthesiology, Baoding First Central Hospital, Baoding, China, 3 Department of Orthopedics, Baoding First Central Hospital, Baoding, China

* 960558692@qq.com

**Data Availability Statement:** All relevant data are within the paper, and all relevant date are available

## Abstract

This study aimed to investigate the minimum number of operations required for itinerant nurses in the operating room to master the skills needed to operate the Ti-robot-assisted spinal surgery equipment. Additionally, we aimed to provide a corresponding basis for the development of qualification admission criteria and skill training for nurses who cooperate with this type of surgery. Nine operating room itinerant nurses independently performed Ti-robot equipment simulations using a spine model as a tool, with 16 operations per trainee. Four evaluation indices were recorded: time spent on equipment preparation and line connections, time spent on image acquisition and transmission, time spent on surgical spine screw placement planning, and time spent on robot arm operation. Individual and general learning curves were plotted using cumulative sum analysis. The number of cases in which the slope of the individual learning curves began to decrease was 3–11 cases, and the number of cases in which the slope of the general learning curve began to decrease was 8 cases. The numbers of cases in which the learning curves began to decrease in the four phases were the 5th, 8th, 11th, and 3rd cases. Itinerant nurses required at least eight cases to master the equipment operation skills of Ti-robot-assisted spinal surgery. Among the four phases, the image acquisition and transmission phases and the surgical spine screw placement planning phase were the most difficult and must be emphasized in future training.

## Introduction

Artificial intelligence (AI) is widely used in the medical industry [1–3]. The development of robotic surgery is aligned with that of minimally invasive and precise surgery [4, 5]. The TINAVI orthopedic robotic assist system (also referred to as Ti-robot) is an orthopedic robot independently developed by the Chinese company TINAVI that can be applied to the spine, limb, pelvis, and other parts of the body during surgery, with the characteristics of reduced intraoperative radiation exposure, accurate positioning, and minimally invasive incision [6, 7].

from the figshare (https://figshare.com). DOI:10.6084/m9.figshare.23497553.

**Funding:** The authors received no specific funding for this work.

The development of robotic surgery has resulted in increased requirements for operating room nurses. Under the current medical conditions in China, equipment manipulation in robot-assisted spinal surgery, including equipment preparation and line connection, image acquisition and transmission, surgical spine screw placement planning under the direction of the surgeon, and execution of robotic arm movements, is often performed by itinerant nurses. Therefore, the proficiency of itinerant nurses in robotic equipment operation greatly impacts the smooth performance of this type of surgery. Learning robot equipment operation skills is a step-by-step learning process of exploration, improvement, enhancement, proficiency, and stability. The cumulative sum analysis (CUSUM) method can quantify the learning process and precisely confirm the level of crossing the learning curve [8, 9].

In this study, the CUSUM method was applied to plot the learning curve of itinerant nurses seeking to master the operation skills of robotic-assisted spinal surgery equipment and to explore the minimum number of operations required for itinerant nurses to master the operating skills of robotic equipment. This provides a reference basis for the future development of qualification access and skill training for operating room nurses to engage in orthopedic robotic surgery.

## Materials and methods

### Ethical approval

Ethical approval was obtained from the Ethics Committee of Baoding No.1 Central Hospital (2021–014), and each trainee signed informed consent.

### Tools used in the research

In this study, we used a fluoroscopic spine mold (Fig 1), a complete orthopedic robotic assist system (Fig 2), and a C-arm X-ray machine. The robotic system was manufactured by the company of TINAVI, China; the model number was TINAVI®, consisting of a robot arm, main control dolly, and navigation camera. The C-arm X-ray machine was manufactured by the company, Siemens AG (model number: ARCADIS Orbic). The study was performed in the robot surgery room of the operating room of the First Central Hospital in Baoding, China.

### Research process

The research began on June 10, 2021, ended on July 30, 2021, and lasted eight weeks. Each experiment was conducted once a week. The inclusion criteria for trainees were bachelor's degree or above; the minimum professional title was junior; ≥5 years of work experience in the operating room; ≥3 years as an itinerant nurse; voluntary participation in this research. Nine operating room nurses provided written informed consent to participate in the study prior to the start of the research. All trainees received theoretical lectures from the Ti-robot manufacturer's engineers and passed a theoretical assessment within one month prior to the start of the research. Each trainee observed the live operation of the manufacturer's engineers at least once through the spine mold demonstration. None of the nine trainees in the research had performed independent operation of the Ti-robot prior to the start of this research. Three male nurses had been involved in orthopedic surgery for a long time with previous operating room nursing experiences.

A project evaluation team was established. The evaluation team included two operating room nurses skilled in operating the Ti-robot and an orthopedic surgeon with a vice-senior title. The evaluation team members set the quality evaluation criteria for the work performed in each phase and informed the trainees (Table 1). The orthopedic surgeon was responsible for

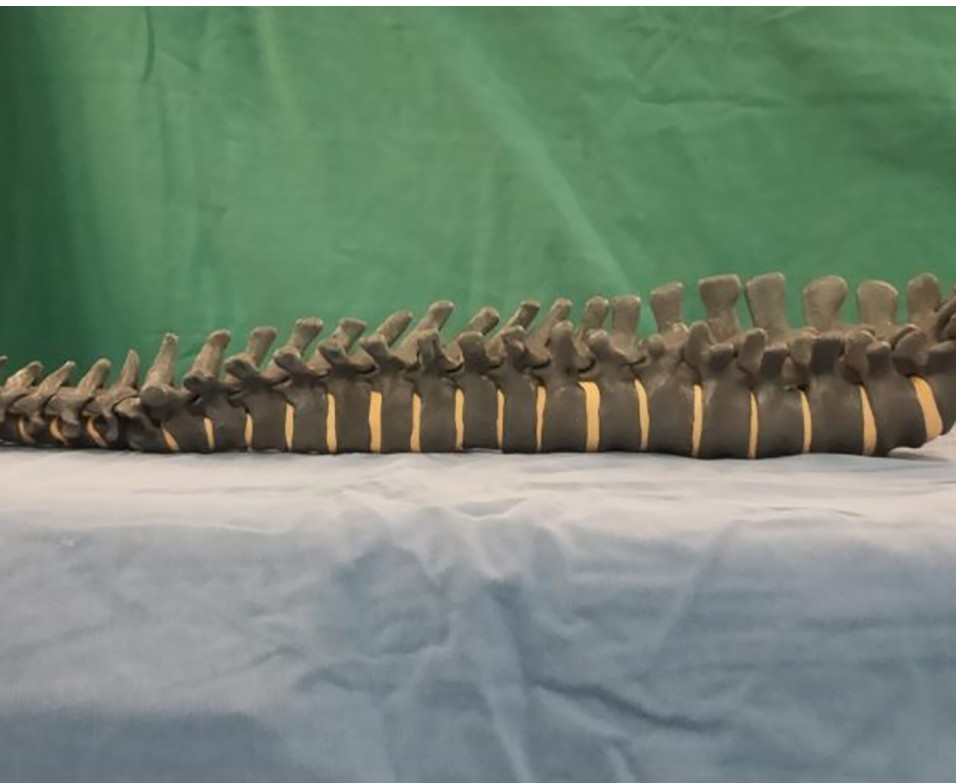

**Fig 1. Fluoroscopic spine mold.**

directing the trainees to perform surgical planning of the corresponding vertebrae in the main control dolly. The trainees performed simulations in the robotic surgery room using the associated equipment and a fluoroscopic spine mold, per the project requirements. The research was conducted for eight weeks, with one day per week selected for the project experiment. In each experiment, each trainee independently performed two complete procedures in the order of the lottery. Each trainee had performed 16 independent operations by the end of the study period. Members of the evaluation team used a stopwatch to record the time spent by each trainee during each phase of the operation.

## Observed indicator

Based on the surgical process of Ti-robot-assisted spinal surgery, the procedure of the trainee operating the robot was divided into four phases. Record the time spent by the trainees in each phase of each operation in the corresponding table. The first phase was equipment preparation and line connection time, which refers to the time taken by the trainees to complete the connection of the C-arm X-ray machine, navigation camera, and main control dolly, in the corresponding order. Additionally, it was necessary to activate the power of each device in the correct order before pushing the corresponding device into the designated preset position. The second phase was the image acquisition and transmission time, which refers to the time taken to complete patient registration and tracer selection at the main control dolly, complete the spinal mold scan using the C-arm X -ray machine in 3D mode, and transmit the acquired images to the main control dolly. The third phase was the surgical spinal screw placement planning time, which refers to the time required to complete the planning of five pairs of spinal

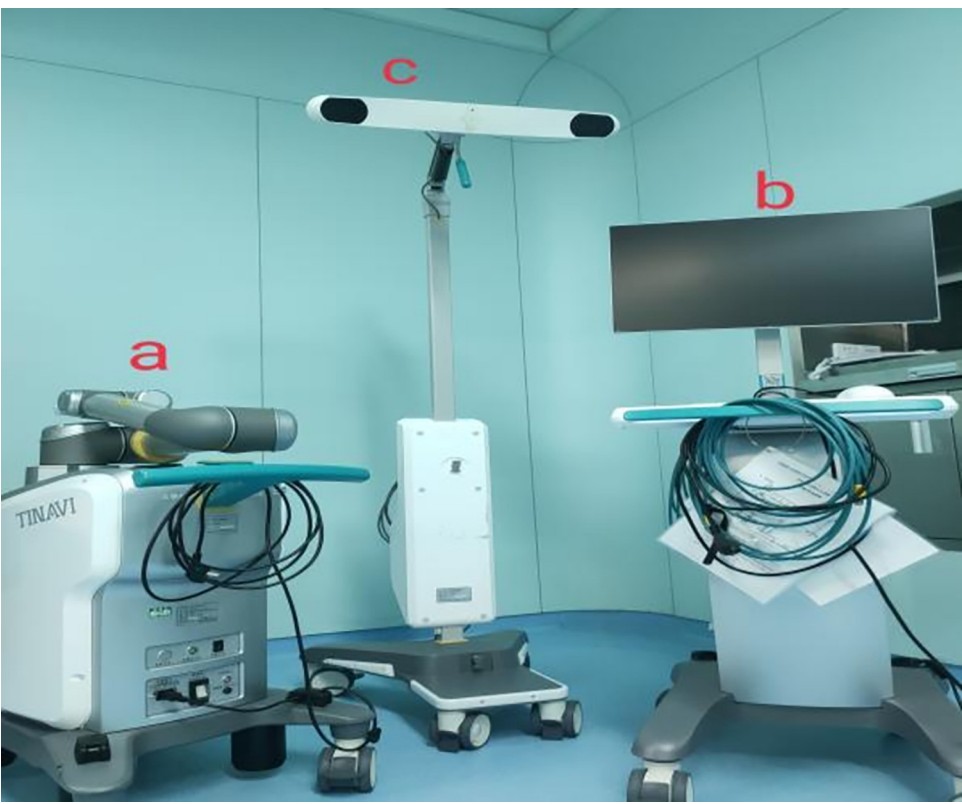

**Fig 2. Ti-orthopedic robotic assist system.** a: robot arm. b: main control dolly. c: navigation camera.

screws from lumbar 1 to lumbar 5 under the direction of the orthopedic surgeon. The fourth phase was the robot arm operation time, which refers to the time taken to run the robot arm to move to the designated position sequentially according to the planning scheme (if the navigation camera could not track the robot arm tracer during the operation, the position of the navigation camera had to be adjusted in a timely manner). The robot arm was withdrawn, and folding and homing were completed.

## Cumulative summation (CUSUM) calculation

The CUSUM method is a time-weighted control chart that calculates the degree of deviation of each sample observation indicator from the target value by summing the CUSUM values over time.

When calculating CUSUM, it is necessary to quantify the observed indicators. The quantitative values of time consumption in the four phases of the trainees were set as $\delta_1$, $\delta_2$, $\delta_3$, $\delta_4$. The formula for calculating the quantized value was: $\delta = X_i - X_0$. $X_0$ was defined as the failure rate of the target value, $X_i$ was defined as whether the target value is reached in each trainee operation, $X_i$ equals 0 when the target value is reached, and $X_i$ equals 1 when the target value is not reached. There are several ways to set the target value: (1) domestic and foreign literature reports; (2) uniform opinions of experts in the relevant specialty areas; (3) baseline data or the average level of skilled operators [10, 11]. After reviewing the literature and manufacturers' training materials, no standard completion time for each operation was found, so the average level of skilled operators in our department was used as the target value in this research. The target values for the four phases were 3, 3, 5, and 3 min. The failure rates of the phases were set

**Table 1. Ti-robot equipment operation record sheet.**

| Serial number | | Operation time | | Number of operations. | |
|---|---|---|---|---|---|
| Operation Phase | Operation standards | | | Standard time (min) | Actual time (min) |
| Phase 1 | **Equipment preparation and line connection phase:** The equipment used are pushed to the designated position. The navigation camera is placed on the head side of the surgical bed. The robot arm is placed on the right side of the surgical bed. The C-arm X-ray machine is pushed from the left side of the surgical bed. The main control dolly is placed at the right front of the head side of the surgical bed. Connect the navigation camera line-the main control dolly line-the DICOM network line-the robot arm power line-the main control dolly power line-the C-arm X-ray power line in turn. After the line connection is completed, turn on the robot arm power-main control dolly power-main control dolly computer power in turn. Login to the general account. Select the appropriate model of robot arm tracer. Set the robot arm preset position to "left scale position." | | | 3 | |
| Phase 2 | **Images acquisition and transmission phase:** Attach the positioning scale to the distal end of the robotic arm tracer after locking the robot arm on the floor. The C-arm X-ray is pushed in from the left side of the surgical bed. The images are acquired in the order of frontal- lateral-3D mode. When acquiring images in the frontal and lateral positions, ensure that all marker points are within the image range. Note that the tracer, robot arm, and mold cannot move during image acquisition; otherwise, the images need to be reacquired. After the image acquisition is successful, select the acquired 3D images from the C-arm X-ray machine console and transfer them to the main control dolly. The transfer process must ensure that the color of the two tracers on the software interface is green, i.e., the tracer is visible to the navigation camera. | | | 3 | |
| Phase 3 | **Surgical spinal screws placement planning phase:** Screws placement planning of the lumbar 1–lumbar 5 vertebrae is performed under the physician's guidance. The selected vertebral body is adjusted to a sagittal and transverse plane suitable for screws planning. When the planning is complete, click "Add Planning" to plan the next vertebra. A total of 10 screws are planned. | | | 5 | |
| Phase 4 | **Robot arm operation phase:** After completing all screws planning, replace the positioning scale at the end of the robot arm with a guide. Before running the screws planning, click "Simulation" to display the motion path, and click "OK" to execute the motion path after confirming the safety. Run to the specified position, fine-tune to an error of less than 0.5 mm and stay 3 seconds before proceeding to the next screw planning path. During the operation of the robot arm, if the tracer cannot be tracked, then there is need to manually adjust the position of the navigation camera until both the robot arm tracer and the mold tracer show green before continuing to run. After all screws planning is completed, withdraw the robot arm. | | | 3 | |

as 5%, 10%, 10%, and 5%, respectively. The quantitative value of each operation for each trainee was the sum of the four component scores. The formula was $\Sigma = \delta 1 + \delta 2 + \delta 3 + \delta 4$.

The cumulative sum formula was $S_i = \sum_{j=1}^{i}(x_j - x_0)$.

The individual cumulative sum, the overall cumulative sum, and the overall cumulative sum of each phase were calculated.

## Statistical analysis

Statistical software (SPSS 20.0) was used to analyze the datasets: the individual learning curve and general learning curve. The general learning curves according to each phase were plotted separately. Scatter plots were created with X-axis representing the number of operation cases, and the Y-axis representing the cumulative sum $S_i$. The curve-fitting method was used to describe the relationships between the curve functions of the X- and Y-axes, and the curve-

**Table 2. Number of operations corresponding to a decrease in the K-value of the individual learning curves of the trainees.**

| Serial number | Age (years) | Gender | Professional Title | Working years | The number of operations corresponding to a decrease in slope K-value |
|---|---|---|---|---|---|
| A | 40 | Female | Intermediate | 18 | 11 |
| B | 29 | Female | Junior | 7 | 9 |
| C | 34 | Male | Intermediate | 10 | 7 |
| D | 31 | Female | Intermediate | 9 | 8 |
| E | 28 | Male | Junior | 5 | 7 |
| F | 27 | Male | Intermediate | 5 | 7 |
| G | 34 | Male | Intermediate | 10 | 3 |
| H | 28 | Female | Junior | 6 | 9 |
| I | 35 | Male | Intermediate | 12 | 6 |

fitting effect was evaluated using the curve-fitting decision coefficient $R^2$. The slope of the learning curve, K-value, was calculated from the curve function, and the X value corresponding to K = 0 was the peak of the curve. The first X value after the curve peak was the minimum number of operations required for the trainees to master the skill.

## Results

The nine trainees had an average age of 31.7 years (range 27–40 years) and included five males and four females, five of whom held an intermediate title and four held junior titles; their average years of operating room work experience was 9.1 years (range: 5–18 years). Before this study, they frequently worked as itinerant nurses and had no experience operating the Ti-robot independently (Table 2). The male nurses numbered C, G, and I, who had been working in orthopedic surgery for a long time, were engaged for 6, 8, and 8 years, respectively, with an average of 7.3 years of work experience.

### Individual learning curves

When the K-value of the learning curve equals zero, the first X value after the peak of the curve is the minimum number of operations required for trainees to master the skill (Fig 3). The number of operations corresponding to a decrease in the K-value of the individual learning curve graph for the nine study participants is shown in Table 2.

### Overall learning curve

The X-axis represents the number of general learning curve operations, and the y-axis represents the cumulative sum of each operation for all trainees. The general learning curve-fitting function equation was Y = 0.0479X³ + 1.6807X² + 16.684X + 20.67, $R^2$ = 0.9326, and the curve fitted well. The slope of this curve showed that at the 7th operation, the K-value was 0. Therefore, eight operations are the minimum number of operations required for trainees to master robotic equipment operation skills (Fig 4).

### Overall learning curve of phases

The X-axis showed the number of operations in each phase of the learning curve, and the Y-axis showed the cumulative sum of operations in each phase for all trainees. The general learning curve fitting function for each phase is shown in Fig 3. The slope of the curve calculated from the fitting function shows that the number of operations in which the slope started to decrease during the four-phase operations were the 5th, 8th, 11th, and 3rd cases (Fig 5).

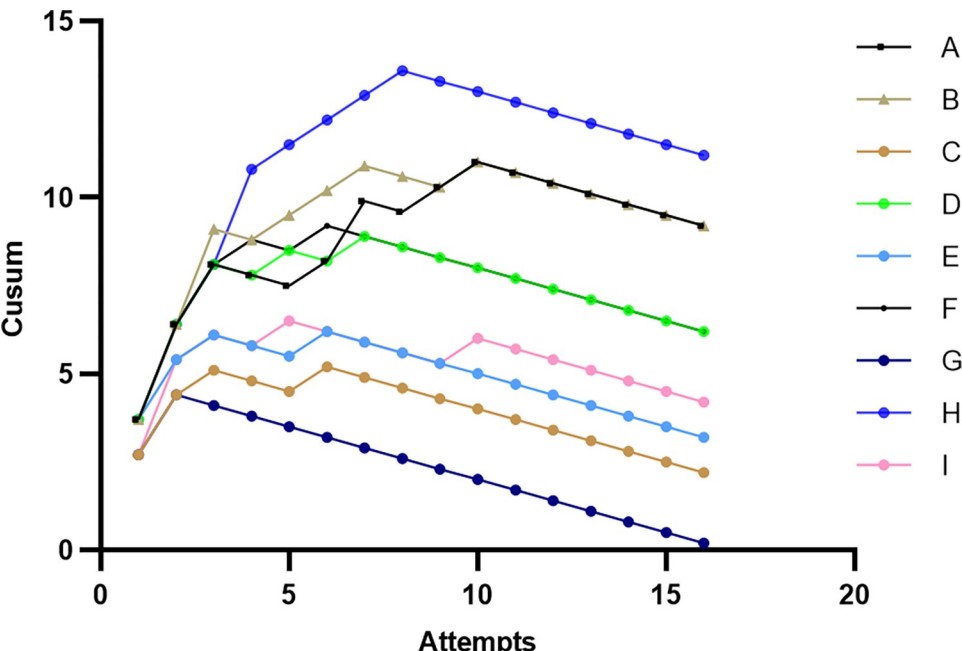

**Fig 3. Cumulative sum chart for Ti-robot operation.** Each color line represents the learning curves of each trainee.

## Discussion

Learning is the process of changing behavior over many repetitions, and it exists for any repetitive manual operation [12, 13]. The CUSUM method, first proposed by Professor *Page* of Cambridge University in 1954, is a mathematical and statistical method recommended in the ISO 9000 standard and is now widely used in research on clinical skills and surgical learning curves [14]. The CUSUM chart can accumulate fluctuations during the learning process to achieve a magnifying effect and detect data points where anomalies occur promptly.

In this study, we found that the number of cases in which the K value started to decrease in the individual learning curve of the nine trainees was at least 3 and at most 11. In the overall learning curve, the number of cases in which the K value began to decline was eight, suggesting that the overall study population was still in the learning and exploring stage for operational skills until the number of operations reached eight, and the trainees required at least eight operations to cross the learning curve and enter a relatively proficient stage. The overall learning curve of each operation stage was also studied, and the number of cases in which the K values of the four stages showed inflection points were the 5th, 9th, 11th, and 3rd cases, indicating that the image acquisition and transmission phases and the surgical spinal screw placement planning phase were the most difficult parts of the learning process.

The study of the individual learning curves of the nine trainees revealed that the minimum number of operations required to become proficient varied considerably between individuals. The number of cases when the learning curve K-value began to decrease in trainee 1 was the 11th case, while the number of cases when the learning curve K-value began to decrease in trainee 7 was the 3rd case. It was also found that female nurses needed more cases to master skills than male nurses (female: 9.25 vs. male: 5). By examining individual learning curves, it was found that the second and third phases, namely, the image acquisition and transmission phases and the surgical spinal screw placement planning phases, were the main phases in which male and female nurses differed. There may be several reasons for this: (1) Female

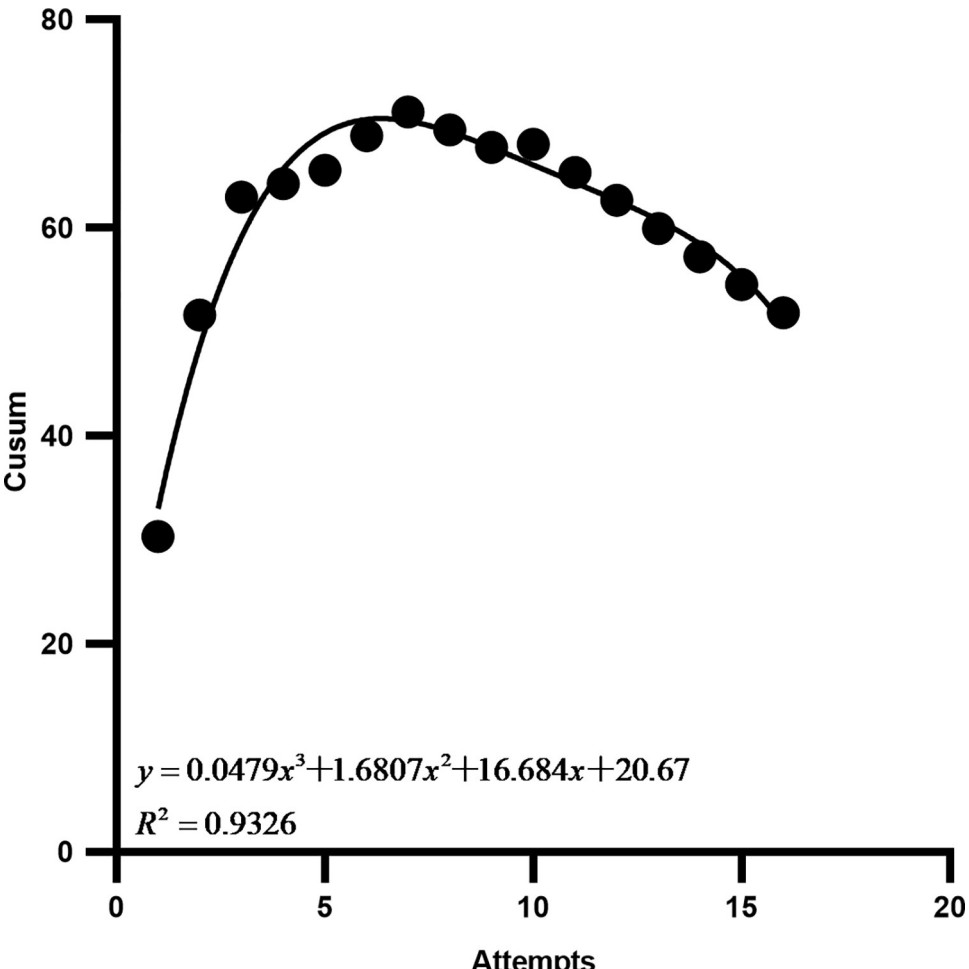

$$y = 0.0479x^3 + 1.6807x^2 + 16.684x + 20.67$$
$$R^2 = 0.9326$$

**Fig 4. General learning curve of 9 trainees.** The curve slope was steadily negative after the 8th operation suggesting that 8 operations crossed the learning curve.

nurses were less powerful than male nurses and took more time to operate large equipment such as C-arm X-ray machines. (2) Male nurses had better acumen than female nurses in understanding and operating robotic equipment. (3) Male nurses had more experience in orthopedic surgery than female nurses and might understand anatomy better than female nurses, thus taking less time in the plane selection of sagittal and transverse sections of the 3D image of the spine and the spine screw placement planning phase when performing surgery in the main control dolly.

Through the study of the overall learning curve, it was found that when the 8th operation was performed, the learning curve K-value began to show a decreasing trend, the curve began to show a flat state, and the learning process showed a stable period, indicating that the trainees had crossed the learning curve and entered a proficient stage. For the overall learning curve of each phase, it was found that the number of cases with inflection points of K-value in the four stages were the 5th, 9th, 11th, and 3rd cases respectively, indicating that the image acquisition and transmission phase and the surgical spinal screws placement planning phase were the most difficult parts to master in the learning process of all stages. Through communication with the study subjects, it was generally found that the difficulties in these two stages were as

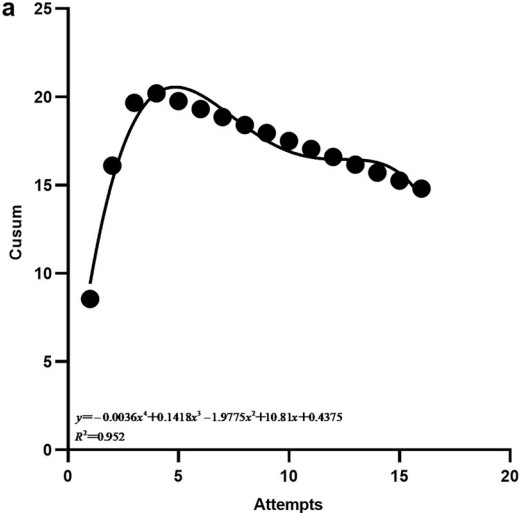

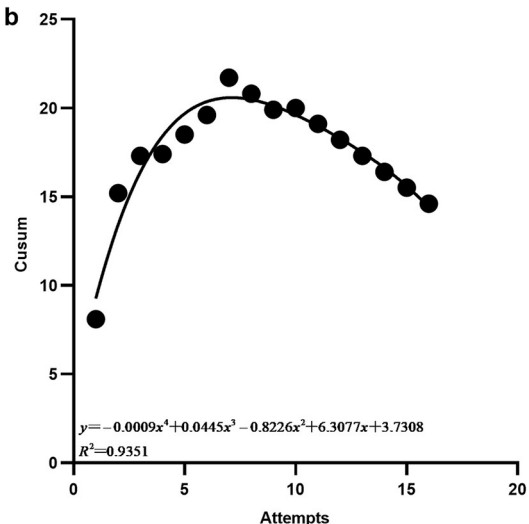

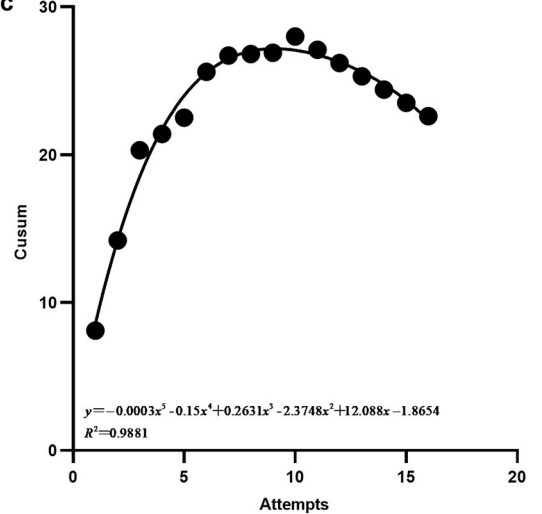

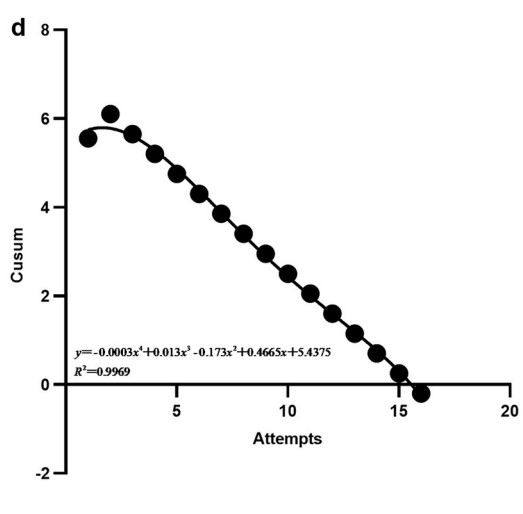

**Fig 5. Cumulative sum chart for Ti-robot operation of phases.** a, b, c, and d represent the overall learning curve of the first, second, third, and fourth stage respectively.

follows: (1) During the acquisition of frontal and lateral images of the spine model using the C-arm X-ray machine, it was necessary to repeatedly adjust the relationship between the positions of the C-arm X-ray machine, spine model, and tracer, making it difficult to succeed in one step. (2) When planning spinal screws on the 3D images received by the main control dolly, it was impossible to quickly and accurately adjust the sagittal and transverse planes

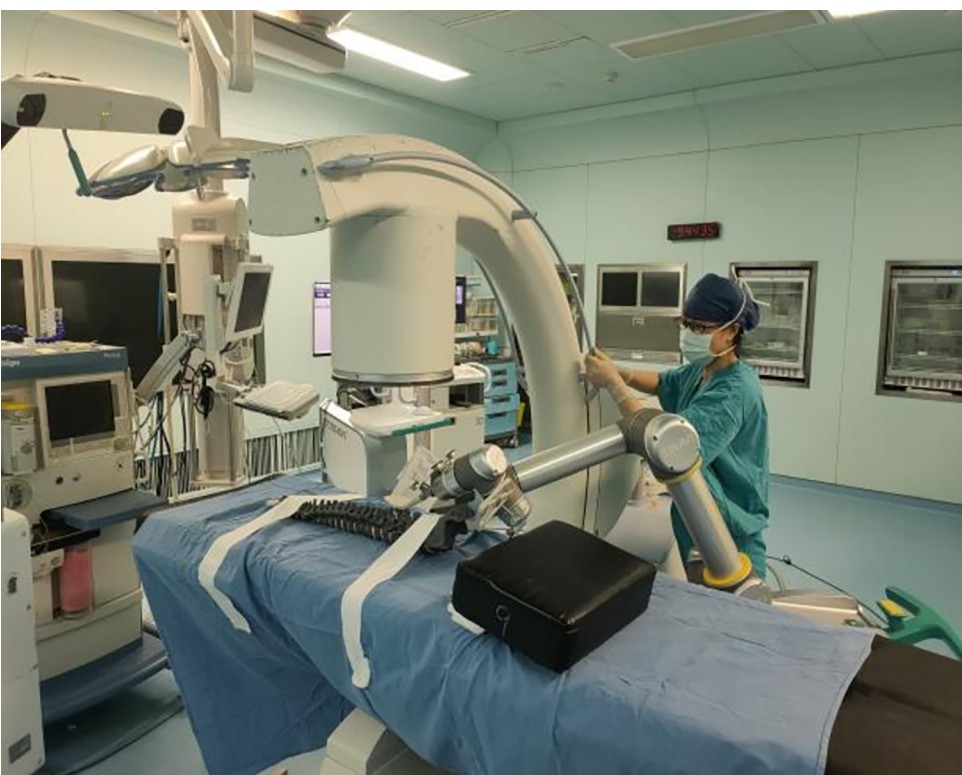

**Fig 6. The trainee capturing the images of spine model.**

suitable for spinal screw planning. (3) When planning spinal screws, the height of screw insertion in the sagittal plane and the angle of screw insertion in the transverse plane must be adjusted repeatedly (Figs 6–8).

Itinerant nurses play an important coordinating role in robotic surgery teamwork, and their changing job descriptions and scope of responsibilities requires addressing additional challenges. However, there is currently no systematic training program for nurses in China regarding Ti robots. A learning curve for itinerant nurses to master the operation of the Ti-robot-assisted spinal surgery equipment has not yet been reported. This is the first study based on spinal molds in China, and the results may provide a basis for future training. Combining the individual and overall learning curves, it is recommended that the itinerant nurse should accumulate at least eight cases of surgical experience and training should focus on both the image acquisition and transmission phases and the surgical nail placement planning phase. Simulation training using molds can simulate special situations that may occur during surgery, increasing the realism of the training and helping trainees' initial learning, while allowing repeated reinforcement exercises for weak links. Operating room managers and robot manufacturer trainers should create supportive training conditions for nurses; in future training, the parts mentioned above can be combined with pictures, videos, and repeated simulation operations to focus on the relevant training to shorten the learning curve of operating room nursing staff [15].

The limitations of this study include its small sample size and single-center nature. In future studies, the sample size could be increased and the study conducted at multiple centers to reduce bias and explore the factors influencing itinerant nurses' mastery of this skill.

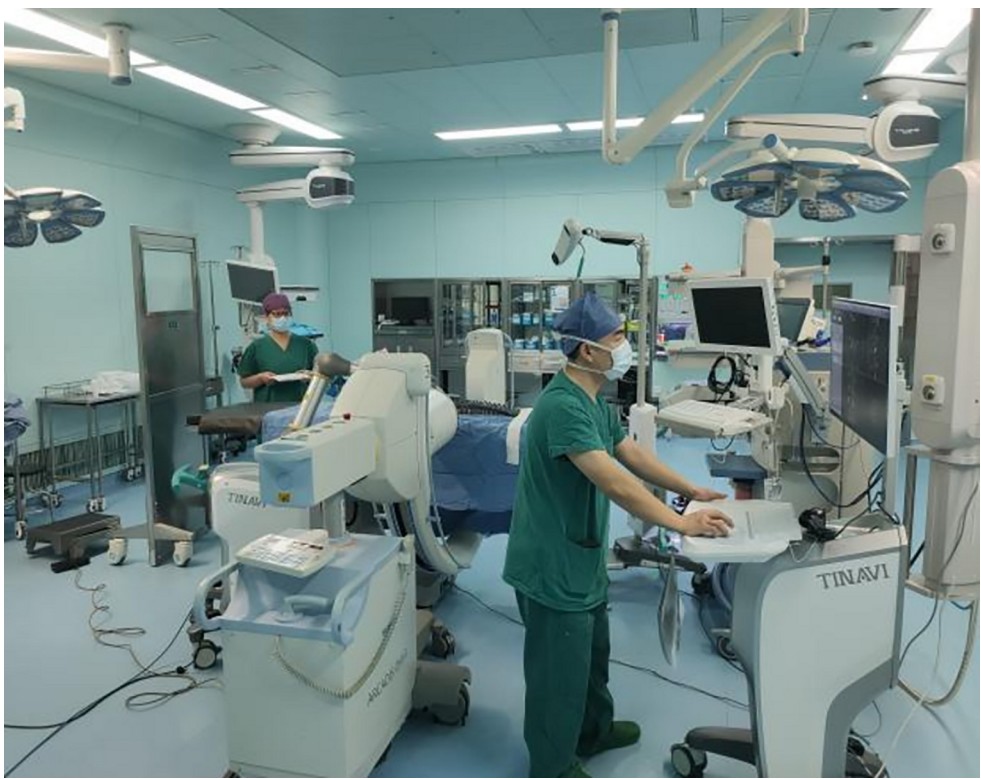

**Fig 7. The trainee planing the spine screws position on main control dolly.**

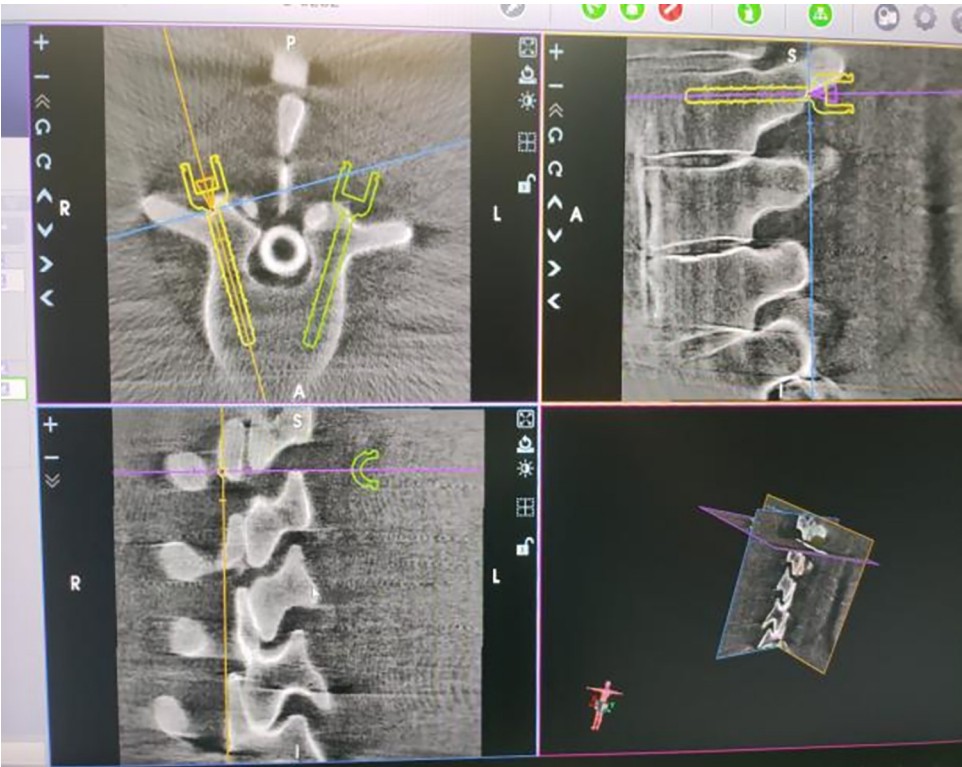

**Fig 8. The sagittal and transverse planes of the images used to plan the height and angle of the screws.**

## Conclusions

Inexperienced itinerant nurses need at least eight operations to become proficient in operating equipment for spinal surgery with the assistance of a Ti-robot.

## Acknowledgments

The authors would like to thank all itinerant nurses who participated in this study.

## Author Contributions

**Conceptualization:** Qi Zhang, Haimao Teng.

**Data curation:** Yichao Yao.

**Formal analysis:** Huiyue Wang, Qi Zhang.

**Investigation:** Qi Zhang, Haimao Teng.

**Methodology:** Huiyue Wang, Hui Qi, Qian Zhang.

**Supervision:** Qian Zhang.

**Writing – original draft:** Yichao Yao.

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
