## [Decision Letter · Decision Letter 0]

29 May 2023

PONE-D-23-11364learning curves for itinerant nurses to master the operation skill of Ti-robot assisted spinal surgery equipmentPLOS ONE

Dear Dr. Yao,

Thank you for submitting your manuscript to PLOS ONE. After careful consideration, we feel that it has merit but does not fully meet PLOS ONE’s publication criteria as it currently stands. Therefore, we invite you to submit a revised version of the manuscript that addresses the points raised during the review process.

We look forward to receiving your revised manuscript.

Kind regards,

Priti Chaudhary, M.S.

Academic Editor

PLOS ONE

Journal Requirements:

  "The research was supported by the Baoding First Central Hospital ,Hebei province,China."

4. Please amend the manuscript submission data (via Edit Submission) to include author Zhang Qi Zhang, Haimao Teng and Hui Qi.

5. Please ensure that you refer to Figure 4 in your text as, if accepted, production will need this reference to link the reader to the figure.

Additional Editor Comments:

Authors are requested to reply all the queries, raised by both the reviewers.

Reviewers' comments:

Reviewer's Responses to Questions

**Comments to the Author**

1. Is the manuscript technically sound, and do the data support the conclusions?

Reviewer #1: Partly

Reviewer #2: Yes

2. Has the statistical analysis been performed appropriately and rigorously? 

Reviewer #1: No

Reviewer #2: Yes

3. Have the authors made all data underlying the findings in their manuscript fully available?

Reviewer #1: Yes

Reviewer #2: Yes

4. Is the manuscript presented in an intelligible fashion and written in standard English?

Reviewer #1: No

Reviewer #2: No

5. Review Comments to the Author

Reviewer #1: First, I would like to express sincere gratitude to get an opportunity to review the manuscript. The endeavor of the authors is appreciated. However, there is some scope for its improvement.

Overall comments:

1. The study design is not clear. Kindly do the needful.

2. The study size is very small.

3. The methods section is poorly framed.

4. The paper needs to be structured scientifically. For example, different sections need to be well framed. Kindly refer to the section-wise concerns herein.

5. It would be better to define primary and secondary outcome variables in methods section in line with objectives. Results, discussion and conclusion need to follow the same in chain.

6. Kindly provide Institutional Review Board reference number and date.

7. The discussion section needs to be described scientifically. Kindly frame it along the following lines:

a. Main findings of the present study

b. Comparison with other studies

c. Implication and explanation of findings

d. Strengths and limitations

e. Conclusion, recommendation and future directions

8. It would be better if study design were clearly mentioned in title, abstract and main text.

9. There are many grammatical errors starting from the title. Kindly do the needful.

Reviewer #2: Overall, this manuscript doesn’t clarify the significance or potential impact of determining the minimum number of operations for itinerant nurses to master Ti-robot assisted spinal surgery equipment. Providing additional details to explain this significance will be helpful. Additionally, there are frequent grammatical errors throughout this manuscript that should be addressed. For example, in section 2, Materials and Methods, "One day was selected for the experiment each week. each trainee signed the informed consent.." These sentences, among others, are not grammatically sound and should be improved.

Certain aspects of the Materials and Methods are also ambiguous, such as "There were three male nurses who have been involved in orthopedic surgery for a long time in previous operating room nursing experiences". It would be helpful to provide more detail into exactly how long these male nurses have been involved in orthopedic surgery.

This is a well executed study that has the potential to be impactful with some edits.

6. PLOS authors have the option to publish the peer review history of their article (what does this mean?). If published, this will include your full peer review and any attached files.

Reviewer #1: **Yes: **Dr Satish Prasad Barnawal

Reviewer #2: No

---

## [Author Response · Author response to Decision Letter 0]

7 Jul 2023

Dear Editor and Reviewers:

 On behalf of my co-authors ,we thank you very much for giving us an opportunity to revise our manuscript ,we appreciate editor and reviewers very much for their positive and constructive comments and suggestions on our manuscript entitled:“learning curves for itinerant nurses to master the operation skill of Ti-robot assisted spinal surgery equipment”.（PONE-D-23-11364）.

 We have studied comments carefully and have made correction which we hope meet with approval. Revised portion are marked in blue and black in the paper. The main corrections in the paper and the responds to the editor and reviewer’s comments are as following:

Dear Editor priti chaudhary:

Responds to the editor’s comments:

Response: Thank you for your notifications.We have carefully checked the style of our manuscript and, as far as possible, revised the style of our manuscript in accordance with PLOS ONE’s request. Maybe some of the details need to be corrected because of the problem of understanding. Please let us know without reservation, and we will continue to improve the style of our manuscript.

2.Please note that funding information should not appear in any section or other areas of your manuscript. We will only publish funding information present in the Funding Statement section of the online submission form. Please remove any funding-related text from the manuscript.

  "The research was supported by the Baoding First Central Hospital ,Hebei province,China."

Response :Thank you for your notifications.We have removed funding information from the manuscript.

Response:Thank you for your notifications.We have uploaded the data we collected during this study to figshare (https://figshare.com)and shared it publicly. DOI:10.6084/m9.figshare 23497553

4.Please amend the manuscript submission data (via Edit Submission) to include author Qi Zhang, Haimao Teng and Hui Qi.

Response : thank you for your notifications.We have refined the information for all authors.

5.Please ensure that you refer to Figure 4 in your text as, if accepted, production will need this reference to link the reader to the figure.

Response: We are sorry to have made such a mistake. This is one of our wrong marks. We have revised it in the manuscript.

Response to Dr Satish Prasad Barnawal

1. The study design is not clear. Kindly do the needful.

Response : Thank you for your notifications. It may be because the research framework is not clear, or it may be because of our language expression problems, so you do not have a clear understanding of our research design. I am deeply sorry for this. Therefore, according to the purpose of our study and the suggestions you have given us, we have made changes to the part of our research framework. At the same time, it also gives guidance to find professional institutions for language problems. Details can be found in our revised manuscript.

2. The study size is very small.

Response : Thank you for your notifications.This is a valuable question that you have raised. Please allow me to explain to you the reason for the relatively small sample size. The use of the Ti- robot has just started in the last few years in the first and second tier cities in China and has not been carried out for a long time. For our hospital, the introduction of the Ti-robot was in December 2019, and in order to get this type of surgery up and running quickly, we only selected a few nurses who had experience in itinerant work as itinerant nurses for the Ti- robot surgery, so this sample size is not very large. This study is the basis and prerequisite for our follow-up study, which is a foundation for subsequent training. This type of study has not been reported in China, and our study may serve as a useful guide for future studies. In future studies, by conducting multi-center studies, the sample size of such studies will increase and more information will be uncovered.

3. The methods section is poorly framed.

Response :Thank you for your notifications. We have made changes to the framework of our methods section as required by the journal. We hope that the changes in language and framework will help you to better understand our research. We also hope that you will provide us with more specific and in-depth comments on the revisions to the methods section.

Here is the framework of the modified method section

Materials and Methods

Ethical approval

Ethical approval was obtained from the Ethics Committee of Baoding No.1 Central Hospital(2021-014), and each trainee signed the informed consent.

Tools used in the research

The tools used in this researchIn this study, we used include a fluoroscopic spine mold (（Fig 1), a complete orthopedic robotic assist system (Fig 2), and a C-arm X-ray machine. The robotic system was manufactured by the company of TINAVI, China; the model number was TINAVIⓇ, consisting of a robot arm, main control dolly, and navigation camera. The C-arm X-ray machine was manufactured by the company, Siemens AG (model number: ARCADIS Orbic). The study was performed in the robot surgery room of the operating room of the First Central Hospital in Baoding, China.

Research process

The research began on June 10, 2021, ended on July 30, 2021, and lasted eight weeks. Each experiment was conducted once a week. The inclusion criteria for trainees were bachelor’s degree or above; the minimum professional title was junior; ≥5 years of work experience in the operating room; ≥3 years as an itinerant nurse; voluntary participation in this research. Nine operating room nurses provided written informed consent to participate in the study prior to the start of the research. All trainees received theoretical lectures from the Ti-robot manufacturer’s engineers and passed a theoretical assessment within one month prior to the start of the research. Each trainee observed the live operation of the manufacturer’s engineers at least once through the spine mold demonstration. None of the nine trainees in the research had performed independent operation of the Ti-robot prior to the start of this research. Three male nurses had been involved in orthopedic surgery for a long time with previous operating room nursing experiences.

A project evaluation team was established. The evaluation team included two operating room nurses skilled in operating the Ti-robot and an orthopedic surgeon with a vice-senior title. The evaluation team members set the quality evaluation criteria for the work performed in each phase and informed the trainees (Table 1). The orthopedic surgeon was responsible for directing the trainees to perform surgical planning of the corresponding vertebrae in the main control dolly. The trainees performed simulations in the robotic surgery room using the associated equipment and a fluoroscopic spine mold, per the project requirements. The research was conducted for eight weeks, with one day per week selected for the project experiment. In each experiment, each trainee independently performed two complete procedures in the order of the lottery. Each trainee had performed 16 independent operations by the end of the study period. Members of the evaluation team used a stopwatch to record the time spent by each trainee during each phase of the operation.

Observed indicator 

Based on the surgical process of Ti-robot-assisted spinal surgery, the procedure of the trainee operating the robot was divided into four phases. Record the time spent by the trainees in each phase of each operation in the corresponding table. The first phase was equipment preparation and line connection time, which refers to the time taken by the trainees to complete the connection of the C-arm X-ray machine, navigation camera, and main control dolly, in the corresponding order. Additionally, it was necessary to activate the power of each device in the correct order before pushing the corresponding device into the designated preset position. The second phase was the image acquisition and transmission time, which refers to the time taken to complete patient registration and tracer selection at the main control dolly, complete the spinal mold scan using the C-arm X -ray machine in 3D mode, and transmit the acquired images to the main control dolly. The third phase was the surgical spinal screw placement planning time, which refers to the time required to complete the planning of five pairs of spinal screws from lumbar 1 to lumbar 5 under the direction of the orthopedic surgeon. The fourth phase was the robot arm operation time, which refers to the time taken to run the robot arm to move to the designated position sequentially according to the planning scheme (if the navigation camera could not track the robot arm tracer during the operation, the position of the navigation camera had to be adjusted in a timely manner). The robot arm was withdrawn, and folding and homing were completed. 

Cumulative summation (CUSUM) calculation

The CUSUM method is a time-weighted control chart that calculates the degree of deviation of each sample observation indicator from the target value by summing the CUSUM values over time. 

When calculating CUSUM, it is necessary to quantify the observed indicators. The quantitative values of time consumption in the four phases of the trainees were set as δ1, δ2, δ3, δ4. The formula for calculating the quantized value was: δ = Xi – X0. X0 was defined as the failure rate of the target value, Xi was defined as whether the target value is reached in each trainee operation, Xi equals 0 when the target value is reached, and Xi equals 1 when the target value is not reached. There are several ways to set the target value: (1) domestic and foreign literature reports; (2) uniform opinions of experts in the relevant specialty areas; (3) baseline data or the average level of skilled operators[10-11]. After reviewing the literature and manufacturers’ training materials, no standard completion time for each operation was found, so the average level of skilled operators in our department was used as the target value in this research. The target values for the four phases were 3, 3, 5, and 3 min. The failure rates of the phases were set as 5%, 10%, 10%, and 5%, respectively. The quantitative value of each operation for each trainee was the sum of the four component scores. The formula was Σ = δ1 + δ2 + δ3 + δ4. The cumulative sum formula was .

The individual cumulative sum, the overall cumulative sum, and the overall cumulative sum of each phase were calculated.

Statistical analysis 

Statistical software (SPSS 20.0) was used to analyze the datasets: the individual learning curve and general learning curve. The general learning curves according to each phase were plotted separately. Scatter plots were created with X-axis representing the number of operation cases, and the Y-axis representing the cumulative sum. The curve-fitting method was used to describe the relationships between the curve functions of the X- and Y-axes, and the curve-fitting effect was evaluated using the curve-fitting decision coefficient R2. The slope of the learning curve, K-value, was calculated from the curve function, and the X value corresponding to K = 0 was the peak of the curve. The first X value after the curve peak was the minimum number of operations required for the trainees to master the skill.

4. The paper needs to be structured scientifically. For example, different sections need to be well framed. Kindly refer to the section-wise concerns herein.

Response :Thank you for your notifications .We have reworked the framing section of our manuscript in accordance with the journal's requirements. The details can be reviewed in the revised manuscript section.

5. It would be better to define primary and secondary outcome variables in methods section in line with objectives. Results, discussion and conclusion need to follow the same in chain.

Response :Thang you for your notifications .In fact, this is not a very complex study. We apologize that we may have used unstandardized language that may have caused you problems in understanding .We have made changes to the method framework section , which may show the relevant variables more clearly compared to the previous version.

Here is the modified variables section

Observed indicator 

Based on the surgical process of Ti-robot-assisted spinal surgery, the procedure of the trainee operating the robot was divided into four phases. Record the time spent by the trainees in each phase of each operation in the corresponding table. The first phase was equipment preparation and line connection time, which refers to the time taken by the trainees to complete the connection of the C-arm X-ray machine, navigation camera, and main control dolly, in the corresponding order. Additionally, it was necessary to activate the power of each device in the correct order before pushing the corresponding device into the designated preset position. The second phase was the image acquisition and transmission time, which refers to the time taken to complete patient registration and tracer selection at the main control dolly, complete the spinal mold scan using the C-arm X -ray machine in 3D mode, and transmit the acquired images to the main control dolly. The third phase was the surgical spinal screw placement planning time, which refers to the time required to complete the planning of five pairs of spinal screws from lumbar 1 to lumbar 5 under the direction of the orthopedic surgeon. The fourth phase was the robot arm operation time, which refers to the time taken to run the robot arm to move to the designated position sequentially according to the planning scheme (if the navigation camera could not track the robot arm tracer during the operation, the position of the navigation camera had to be adjusted in a timely manner). The robot arm was withdrawn, and folding and homing were completed. 

Cumulative summation (CUSUM) calculation

The CUSUM method is a time-weighted control chart that calculates the degree of deviation of each sample observation indicator from the target value by summing the CUSUM values over time. 

When calculating CUSUM, it is necessary to quantify the observed indicators. The quantitative values of time consumption in the four phases of the trainees were set as δ1, δ2, δ3, δ4. The formula for calculating the quantized value was: δ = Xi – X0. X0 was defined as the failure rate of the target value, Xi was defined as whether the target value is reached in each trainee operation, Xi equals 0 when the target value is reached, and Xi equals 1 when the target value is not reached. There are several ways to set the target value: (1) domestic and foreign literature reports; (2) uniform opinions of experts in the relevant specialty areas; (3) baseline data or the average level of skilled operators[10-11]. After reviewing the literature and manufacturers’ training materials, no standard completion time for each operation was found, so the average level of skilled operators in our department was used as the target value in this research. The target values for the four phases were 3, 3, 5, and 3 min. The failure rates of the phases were set as 5%, 10%, 10%, and 5%, respectively. The quantitative value of each operation for each trainee was the sum of the four component scores. The formula was Σ = δ1 + δ2 + δ3 + δ4. The cumulative sum formula was .

The individual cumulative sum, the overall cumulative sum, and the overall cumulative sum of each phase were calculated.

6. Kindly provide Institutional Review Board reference number and date.

Response:Thank you for the heads up. We have added the ethics approval number to the manuscript and have uploaded the ethics approval as required by PLOS ONE.

We added the ethics approval number to line 75 of the manuscript.

Ethical approval was obtained from the Ethics Committee of Baoding No.1 Central Hospital[2021-014], and each trainee signed the informed consent.

7. The discussion section needs to be described scientifically. Kindly frame it along the following lines:

a. Main findings of the present study

b. Comparison with other studies

c. Implication and explanation of findings

d. Strengths and limitations

e. Conclusion, recommendation and future directions

Response:

Thank you for your notifications.

Based on your suggestion, in the second paragraph of the Discussion section, we mainly present the results of our study. In the third and fourth paragraphs of the Discussion section, we describe the implications and interpretation of the study results. In the fifth and sixth paragraphs of the discussion section, we describe the strengths and limitations of this study, as well as the recommendations and directions of this study for future research.

Our team members have not found any similar studies in China for the time being by reviewing the literature. Therefore, at this stage, we believe that there is no feasibility of comparison with existing studies. It is possible that as the study is conducted, more and more similar studies will be conducted and we will keep following up the relevant studies.

The following is part of our improved discussion.

Learning is the process of changing behavior over many repetitions, and it exists for any repetitive manual operation [12-13]. The CUSUM method, first proposed by Professor Page of Cambridge University in 1954, is a mathematical and statistical method recommended in the ISO 9000 standard and is now widely used in research on clinical skills and surgical learning curves [14]. The CUSUM chart can accumulate fluctuations during the learning process to achieve a magnifying effect and detect data points where anomalies occur promptly. 

In this study, we found that the number of cases in which the K value started to decrease in the individual learning curve of the nine trainees was at least 3 and at most 11. In the overall learning curve, the number of cases in which the K value began to decline was eight, suggesting that the overall study population was still in the learning and exploring stage for operational skills until the number of operations reached eight, and the trainees required at least eight operations to cross the learning curve and enter a relatively proficient stage. The overall learning curve of each operation stage was also studied, and the number of cases in which the K values of the four stages showed inflection points were the 5th, 9th, 11th, and 3rd cases, indicating that the image acquisition and transmission phases and the surgical spinal screw placement planning phase were the most difficult parts of the learning process.

The study of the individual learning curves of the nine trainees revealed that the minimum number of operations required to become proficient varied considerably between individuals. The number of cases when the learning curve K-value began to decrease in trainee 1 was the 11th case, while the number of cases when the learning curve K-value began to decrease in trainee 7 was the 3rd case. It was also found that female nurses needed more cases to master skills than male nurses (female: 9.25 vs. male: 5). By examining individual learning curves, it was found that the second and third phases, namely, the image acquisition and transmission phases and the surgical spinal screw placement planning phases, were the main phases in which male and female nurses differed. There may be several reasons for this: (1) Female nurses were less powerful than male nurses and took more time to operate large equipment such as C-arm X-ray machines. (2) Male nurses had better acumen than female nurses in understanding and operating robotic equipment. (3) Male nurses had more experience in orthopedic surgery than female nurses and might understand anatomy better than female nurses, thus taking less time in the plane selection of sagittal and transverse sections of the 3D image of the spine and the spine screw placement planning phase when performing surgery in the main control dolly.

Through the study of the general learning curve, it was found that when the 8th operation was performed, the learning curve K-value began to show a decreasing trend, the curve began to show a flat state, and the learning process showed a stable period, indicating that the trainees had crossed the learning curve and entered a proficient stage. For the general learning curve of each phase, it was found that the number of cases with inflection points of K-value in the four stages were the 5th, 9th, 11th, and 3rd cases respectively, indicating that the image acquisition and transmission phase and the surgical spinal screws placement planning phase were the most difficult parts to master in the learning process of all stages. Through communication with the study subjects, it was generally found that the difficulties in these two stages were as follows: (1) During the acquisition of frontal and lateral images of the spine model using the C-arm X-ray machine, it was necessary to repeatedly adjust the relationship between the positions of the C-arm X-ray machine, spine model, and tracer, making it difficult to succeed in one step. (2) When planning spinal screws on the 3D images received by the main control dolly, it was impossible to quickly and accurately adjust the sagittal and transverse planes suitable for spinal screw planning. (3) When planning spinal screws, the height of screw insertion in the sagittal plane and the angle of screw insertion in the transverse plane must be adjusted repeatedly. (Figs 6–8).

Itinerant nurses play an important coordinating role in robotic surgery teamwork, and their changing job descriptions and scope of responsibilities requires addressing additional challenges. However, there is currently no systematic training program for nurses in China regarding Ti robots. A learning curve for itinerant nurses to master the operation of the Ti-robot-assisted spinal surgery equipment has not yet been reported. This is the first study based on spinal molds in China, and the results may provide a basis for future training. Combining the individual and overall learning curves, it is recommended that the itinerant nurse should accumulate at least eight cases of surgical experience and training should focus on both the image acquisition and transmission phases and the surgical nail placement planning phase. Simulation training using molds can simulate special situations that may occur during surgery, increasing the realism of the training and helping trainees' initial learning, while allowing repeated reinforcement exercises for weak links. Operating room managers and robot manufacturer trainers should create supportive training conditions for nurses; in future training, the parts mentioned above can be combined with pictures, videos, and repeated simulation operations to focus on the relevant training to shorten the learning curve of operating room nursing staff [15].

The limitations of this study include its small sample size and single-center nature. In future studies, the sample size could be increased and the study conducted at multiple centers to reduce bias and explore the factors influencing itinerant nurses' mastery of this skill.

8. It would be better if study design were clearly mentioned in title, abstract and main text.

Response :Thank you for you notifications .The current study focused on the learning curve of itinerant nurses in mastering the operation of the Ti- robotic device using the CUSUM method. We have modified the title section and also referred to the CUSUM method in the abstract section and the body section. By making changes to the framework and language, we hope to make our study design clearer to you and the reader.

9. There are many grammatical errors starting from the title. Kindly do the needful.

Response:Thank you for your critical comments. We are sorry to have made such a mistake. We revised the whole manuscript carefully to avoid language errors. In addition,we consulted a professional editing service and asked several colleagues who are native English speakers to check the English. We hope that the language is now acceptable for the review process.

Response to the Review 2

1.Overall, this manuscript doesn’t clarify the significance or potential impact of determining the minimum number of operations for itinerant nurses to master Ti-robot assisted spinal surgery equipment. Providing additional details to explain this significance will be helpful. Additionally, there are frequent grammatical errors throughout this manuscript that should be addressed. For example, in section 2, Materials and Methods, "One day was selected for the experiment each week. each trainee signed the informed consent.." These sentences, among others, are not grammatically sound and should be improved.

Response : Thank you for your notifications. Perhaps because our language and grammar are not standardized, Let the content of our manuscript not be presented clearly.We revised the whole manuscript carefully to avoid language errors. In addition,we consulted a professional editing service and asked several colleagues who are native English speakers to check the English. We hope that you can review our research. The sample size of this study is not large, please allow me to explain to you the reason for the relatively small sample size. The use of the Ti- robot has just started in the last few years in the first and second tier cities in China and has not been carried out for a long time. For our hospital, the introduction of the Ti-robot was in December 2019, and in order to get this type of surgery up and running quickly, we only selected a few nurses who had experience in itinerant work as itinerant nurses for the Ti- robot surgery, so this sample size is not very large. This study is the basis and prerequisite for our follow-up study, which is a foundation for subsequent training. Its significance lies in that it is the first study in China on the learning curve of circuit nurses in mastering the operation of Ti-robot equipment, and it has certain reference significance for relevant institutions to carry out the training of Ti-robot equipment in the future. For example, we suggest that itinerant nurses should undergo at least 8 times of simulation training before they can independently cooperate with Ti-robot assisted spinal surgery. We found that the image acquisition and transmission phase, as well as the screw planning phase, were the difficulties of itinerant nurses, and should be focused on training in the future.

2.Certain aspects of the Materials and Methods are also ambiguous, such as "There were three male nurses who have been involved in orthopedic surgery for a long time in previous operating room nursing experiences". It would be helpful to provide more detail into exactly how long these male nurses have been involved in orthopedic surgery.

Response :Thank you for your notifications .We have re-edited the framework of the Materials and Methods section, hoping to present this section more clearly. For your question, we have provided more details to describe. We added the duration of the three male nurses involved in orthopedic surgery in lines 180 to 183 in the manuscript section(The male nurses numbered C, G, and I, who had been working in orthopedic surgery for a long time, were engaged for 6, 8, and 8 years, respectively, with an average of 7.3 years of work experience.). However, we consider that due to the small sample size, this study cannot provide more data to explore the relevant influencing factors at present. This is one of the questions that we need to address in the next step of conducting a multicenter study to expand the sample size.

Here is the framework of the modified method section:

Materials and Methods

Ethical approval

Ethical approval was obtained from the Ethics Committee of Baoding No.1 Central Hospital(2021-014), and each trainee signed the informed consent.

Tools used in the research

The tools used in this researchIn this study, we used include a fluoroscopic spine mold (（Fig 1), a complete orthopedic robotic assist system (Fig 2), and a C-arm X-ray machine. The robotic system was manufactured by the company of TINAVI, China; the model number was TINAVIⓇ, consisting of a robot arm, main control dolly, and navigation camera. The C-arm X-ray machine was manufactured by the company, Siemens AG (model number: ARCADIS Orbic). The study was performed in the robot surgery room of the operating room of the First Central Hospital in Baoding, China.

Research process

The research began on June 10, 2021, ended on July 30, 2021, and lasted eight weeks. Each experiment was conducted once a week. The inclusion criteria for trainees were bachelor’s degree or above; the minimum professional title was junior; ≥5 years of work experience in the operating room; ≥3 years as an itinerant nurse; voluntary participation in this research. Nine operating room nurses provided written informed consent to participate in the study prior to the start of the research. All trainees received theoretical lectures from the Ti-robot manufacturer’s engineers and passed a theoretical assessment within one month prior to the start of the research. Each trainee observed the live operation of the manufacturer’s engineers at least once through the spine mold demonstration. None of the nine trainees in the research had performed independent operation of the Ti-robot prior to the start of this research. Three male nurses had been involved in orthopedic surgery for a long time with previous operating room nursing experiences.

A project evaluation team was established. The evaluation team included two operating room nurses skilled in operating the Ti-robot and an orthopedic surgeon with a vice-senior title. The evaluation team members set the quality evaluation criteria for the work performed in each phase and informed the trainees (Table 1). The orthopedic surgeon was responsible for directing the trainees to perform surgical planning of the corresponding vertebrae in the main control dolly. The trainees performed simulations in the robotic surgery room using the associated equipment and a fluoroscopic spine mold, per the project requirements. The research was conducted for eight weeks, with one day per week selected for the project experiment. In each experiment, each trainee independently performed two complete procedures in the order of the lottery. Each trainee had performed 16 independent operations by the end of the study period. Members of the evaluation team used a stopwatch to record the time spent by each trainee during each phase of the operation.

Observed indicator 

Based on the surgical process of Ti-robot-assisted spinal surgery, the procedure of the trainee operating the robot was divided into four phases. Record the time spent by the trainees in each phase of each operation in the corresponding table. The first phase was equipment preparation and line connection time, which refers to the time taken by the trainees to complete the connection of the C-arm X-ray machine, navigation camera, and main control dolly, in the corresponding order. Additionally, it was necessary to activate the power of each device in the correct order before pushing the corresponding device into the designated preset position. The second phase was the image acquisition and transmission time, which refers to the time taken to complete patient registration and tracer selection at the main control dolly, complete the spinal mold scan using the C-arm X -ray machine in 3D mode, and transmit the acquired images to the main control dolly. The third phase was the surgical spinal screw placement planning time, which refers to the time required to complete the planning of five pairs of spinal screws from lumbar 1 to lumbar 5 under the direction of the orthopedic surgeon. The fourth phase was the robot arm operation time, which refers to the time taken to run the robot arm to move to the designated position sequentially according to the planning scheme (if the navigation camera could not track the robot arm tracer during the operation, the position of the navigation camera had to be adjusted in a timely manner). The robot arm was withdrawn, and folding and homing were completed. 

Cumulative summation (CUSUM) calculation

The CUSUM method is a time-weighted control chart that calculates the degree of deviation of each sample observation indicator from the target value by summing the CUSUM values over time. 

When calculating CUSUM, it is necessary to quantify the observed indicators. The quantitative values of time consumption in the four phases of the trainees were set as δ1, δ2, δ3, δ4. The formula for calculating the quantized value was: δ = Xi – X0. X0 was defined as the failure rate of the target value, Xi was defined as whether the target value is reached in each trainee operation, Xi equals 0 when the target value is reached, and Xi equals 1 when the target value is not reached. There are several ways to set the target value: (1) domestic and foreign literature reports; (2) uniform opinions of experts in the relevant specialty areas; (3) baseline data or the average level of skilled operators[10-11]. After reviewing the literature and manufacturers’ training materials, no standard completion time for each operation was found, so the average level of skilled operators in our department was used as the target value in this research. The target values for the four phases were 3, 3, 5, and 3 min. The failure rates of the phases were set as 5%, 10%, 10%, and 5%, respectively. The quantitative value of each operation for each trainee was the sum of the four component scores. The formula was Σ = δ1 + δ2 + δ3 + δ4. The cumulative sum formula was .

The individual cumulative sum, the overall cumulative sum, and the overall cumulative sum of each phase were calculated.

Statistical analysis 

Statistical software (SPSS 20.0) was used to analyze the datasets: the individual learning curve and general learning curve. The general learning curves according to each phase were plotted separately. Scatter plots were created with X-axis representing the number of operation cases, and the Y-axis representing the cumulative sum. The curve-fitting method was used to describe the relationships between the curve functions of the X- and Y-axes, and the curve-fitting effect was evaluated using the curve-fitting decision coefficient R2. The slope of the learning curve, K-value, was calculated from the curve function, and the X value corresponding to K = 0 was the peak of the curve. The first X value after the curve peak was the minimum number of operations required for the trainees to master the skill.

2.This is a well executed study that has the potential to be impactful with some edits.

Response :Although our manuscript, there are still some problems. However, we can see your positive evaluation of us, and we are very grateful for your appreciation.

Thanks again to all editors and reviewers for their comments on revisions. We look forward to hearing from you

Best wishes 

Yichao Yao June 27, 2023



---

## [Editor Report · Decision Letter 1]

31 Jul 2023

PONE-D-23-11364R1Learning curves for itinerant nurses to master the operation skill of Ti-robot-assisted spinal surgery equipment by CUSUM analysisPLOS ONE

Dear Dr. Yao,

Thank you for submitting your manuscript to PLOS ONE. After careful consideration, we feel that it has merit but does not fully meet PLOS ONE’s publication criteria as it currently stands. Therefore, we invite you to submit a revised version of the manuscript that addresses the points raised during the review process.

ACADEMIC EDITOR: 

1. Authors are requested to write Funding information in Financial Disclosure column (as per authors-"The research was supported by the Baoding First Central Hospital ,Hebei province,China.")

2. As the sample size is small, study may be considered as Pilot project/ study.

We look forward to receiving your revised manuscript.

Kind regards,

Priti Chaudhary, M.S.

Academic Editor

PLOS ONE

Journal Requirements:

Additional Editor Comments:

1. Authors are requested to write Funding information in Financial Disclosure column (as per authors-"The research was supported by the Baoding First Central Hospital ,Hebei province,China.")

2. As the sample size is small, study may be considered as Pilot project/ study.

---

## [Author Response · Author response to Decision Letter 1]

7 Aug 2023

Dear Editor and Reviewers:

 We are very glad to receive your reply.On behalf of my co-authors ,we thank you very much for giving us an opportunity to revise our manuscript again ,we appreciate editor and reviewers very much for their positive and constructive comments and suggestions on our manuscript entitled:“learning curves for itinerant nurses to master the operation skill of Ti-robot assisted spinal surgery equipment by CUSUM analysis”.（PONE-D-23-11364R1）.

 We have studied comments carefully and have made correction which we hope meet with approval. Revised portion are marked in red in the paper. The main corrections in the paper and the responds to the editor and reviewer’s comments are as following:

Dear Editor priti chaudhary:

Responds to the editor’s comments:

1. Authors are requested to write Funding information in Financial Disclosure column (as per authors-"The research was supported by the Baoding First Central Hospital ,Hebei province,China.")

Response: Thank you for your notifications.We have added Funding information in the corresponding sections.But we have some questions about why this part is not displayed in the generated PDF file? Looking forward to your reply.

2. As the sample size is small, study may be considered as Pilot project/ study.

Response :Thank you for your suggestion.We think this is a very good proposal.We have revised the title of the article.The revised title is “Learning curves for itinerant nurses to master the operation skill of Ti-robot-assisted spinal surgery equipment by CUSUM analysis：A pilot study”

In addition, we added a co first author Huiyue Wang, who contributed substantially to the statistical portion of the study.

 Following the request of the journal, we modified the format of the references.

According to your prompts, we have uploaded our figure files to the PACE digital diagnostic tool, ensured that figures meet PLOS requirements.

Thanks again to all editors and reviewers for their comments on revisions. We look forward to hearing from you.

 Best wishes 

Yichao Yao August 5, 2023

---

## [Editor Report · Decision Letter 2]

23 Aug 2023

Learning curves for itinerant nurses to master the operation skill of Ti-robot-assisted spinal surgery equipment by CUSUM analysis:A pilot study

PONE-D-23-11364R2

Dear Dr. Yi Chao Yao,

We’re pleased to inform you that your manuscript has been judged scientifically suitable for publication and will be formally accepted for publication once it meets all outstanding technical requirements.

Kind regards,

Priti Chaudhary, M.S.

Academic Editor

PLOS ONE
---

## [Editor Report · Acceptance letter]

28 Feb 2024

PONE-D-23-11364R2 

PLOS ONE

Dear Dr. Yao, 

I'm pleased to inform you that your manuscript has been deemed suitable for publication in PLOS ONE. Congratulations! Your manuscript is now being handed over to our production team.

Kind regards, 

on behalf of

Dr. Priti Chaudhary 

Academic Editor

PLOS ONE